# Using language in social media posts to study the network dynamics of depression longitudinally

Sean W. Kelley [1,2✉] & Claire M. Gillan [1,2,3✉]

Network theory of mental illness posits that causal interactions between symptoms give rise to mental health disorders. Increasing evidence suggests that depression network connectivity may be a risk factor for transitioning and sustaining a depressive state. Here we analysed social media (Twitter) data from 946 participants who retrospectively self-reported the dates of any depressive episodes in the past 12 months and current depressive symptom severity. We construct personalised, within-subject, networks based on depression-related linguistic features. We show an association existed between current depression severity and 8 out of 9 text features examined. Individuals with greater depression severity had higher overall network connectivity between depression-relevant linguistic features than those with lesser severity. We observed within-subject changes in overall network connectivity associated with the dates of a self-reported depressive episode. The connectivity within personalized networks of depression-associated linguistic features may change dynamically with changes in current depression symptoms.

[1] School of Psychology, Trinity College Dublin, Dublin, Ireland. [2] Trinity College Institute of Neuroscience, Trinity College Dublin, Dublin, Ireland. [3] Global Brain Health Institute, Trinity College Dublin, Dublin, Ireland. ✉email: sekelley@tcd.ie; gillancl@tcd.ie

Network theories of mental illness propose that disorders like depression emerge from cascades of casual interactions that occur between symptoms[1]. In contrast to traditional frameworks that suggest symptoms are indicators of a single underlying disease state, network theories posit that these symptoms and their interactions are actually what drive these conditions. For example, diminished feelings of worth, compounded by insomnia, may lead to a loss of energy, resulting in weight gain and a decreased ability to think and concentrate. Positive feedback among these symptoms is thought to contribute to the maintenance of a depressive episode[2,3].

Preliminary support for network theory has come from studies comparing the network structure of self-reported depressive symptoms between groups of individuals with and without a diagnosis, or before and after some intervention. Individuals with depression, compared to individuals without depression, are thought to have greater connectivity between depression symptoms, reflecting an elevated vulnerability to 'knock-on effects', that may result in fairly sudden and persistent changes in depression[2]. This has been partially born out in the data; studies have shown that patients with depression have increased connectivity among depression symptoms compared to healthy controls[4–6] and the same is true for several other mental health conditions[7–10]. Moreover, participants who go on to have persistent depression have more strongly connected networks at baseline than those who later enter remission[11,12] and the same appears to be true for patients with eating disorders who undergo treatment[13]. As a change in a system's state approaches, e.g., onset of a depressive episode, network connectivity is expected to increase, reflecting elevated vulnerability, as the system becomes less and less able to recover from external stress[14–16]. One study found some evidence that network connectivity increases approaching a depressive episode, although these effects did not directly capture the onset of an episode and were not shown within-subject[17].

Despite these results, recent studies have yielded some inconsistent findings. Another study failed to extend the findings of Van Borkulo and colleagues (2015) to an adolescent sample—finding instead no difference in baseline network connectivity in patients who went on to have worse outcomes[18]. Furthermore, several studies have failed to find evidence for one of the key predictions of network theory, that individuals who recover (e.g., following treatment) show reductions in their network connectivity. In fact, several studies have actually shown an increase in network connectivity following treatment for depression[12,19–21].

One explanation for the lack of consistent findings is that the majority of prior work has been based on networks constructed from group-level symptom correlations (i.e. between-subject, cross-sectional). An alternative approach is to construct networks based on repeated assessments gathered from the same individual over time (i.e. within subject, longitudinal), which allows one to characterize each individual's network structure, sometimes referred to as a personalised network[22]. This may be an important distinction, because it is unclear to what extent cross-sectional networks capture how symptoms causally relate to one-another over time, within an individual. This issue was addressed empirically when researchers analysed the same dataset in multiple ways, allowing them to directly compare the structure of cross-sectional networks versus personalized ones[23]. They found that the two analysis approaches can sometimes yield different results, including different associations between symptoms and finding that different symptoms were the most central.

We therefore considered if the inconsistent findings in the field of network analysis and mental health to date might stem from the over-reliance on cross-sectional methods for characterizing network structure. Longitudinal studies are needed to test some of the key predictions of network theory, such as whether network connectivity increases as someone transitions from a healthy to acutely ill state. We know of only two network studies that attempted to measure symptoms within the same individual over a long enough period to capture a naturally occurring change in mental health state. These two examples are both single-subject observational studies. Wichers et al. (2016) reported data from a single patient over 239 days, a period of time that spanned the transition into a depressed state, concluding that the patient's network connectivity increased prior to the onset of the depressive episode, though data were insufficient for a formal analysis[24]. In a longitudinal study of one patient with psychosis, researchers similarly observed a qualitative increase in network connectivity during both an impending and full relapse[25]. While these studies are suggestive, larger samples and formal statistical tests are needed to address one of the most fundamental predictions of network theory of mental illness - does network connectivity increase during a depressive episode? The present study aimed to fill this gap by comparing the connectivity of networks within versus outside a depressive episode. Rather than using self-report symptoms, however, we used linguistic features associated with depression posted by users on the social media platform Twitter[26–29].

One reason that studies addressing this question are lacking is because the data required is challenging to gather; multiple assessments are required per day, per participant, over several weeks or months. Challenges are compounded by the need for a naturally occurring depressive episode to have its onset during this period. To circumvent these challenges, we utilized an alternative to ecological momentary assessment (EMA). Instead of asking participants to report their mood, motivation, sleep etc. daily over a prolonged period of time, we analysed depression-associated textual data already archived on Twitter, a social media platform. We analysed data over a 12-month period from nearly 1,000 participants that in some cases spanned the onset of a depressive episode. Central to our approach are prior observations that individuals with depression, both people with a clinical diagnosis and those with self-reported depression symptoms, have significant linguistic differences in both their writing and speech patterns compared to those without depression. For example, individuals with depression use significantly more 1st person singular pronouns than individuals without depression in personal essays[30,31] and semi-structured interviews[32], which is thought to reflect enhanced self-focused attention that occurs in a depressed state[27]. Along with changes in pronoun use, depression is associated with negatively biased cognitive distortions, e.g., everyone thinks that I am a loser[33]. These findings are also observable in social media posts[26,28,34,35], and include increased use of swear and negation words, anger, references to death, and changes in the use of articles and other pronouns[36,37]. People with depression are also less active on Twitter in the early morning (3am–6am) than healthy controls exhibiting an altered circadian rhythm, but also used significantly more personal pronouns, negative affect words, and rumination words during this time. Language usage on social media is thus not static, instead changing over time reflecting underlying changes in a participant's mental health[38]. By examining longitudinal data, we can examine fluctuations in time within-subjects, allowing us to ask if these depression-associated linguistic features become more connected when someone is in the midst of a depressive episode. For example, when a person is more self-focused, using words like "I", "me" and "my", is that person also more anxious, angry or sad? Is that association stronger when someone is currently depressed than when they feel well? This sort of data has the benefit of being objective and relatively plentiful, but the mapping of these linguistic features onto to specific self-reported symptoms diagnostic of depression (such as sadness, sleep disturbances

**Table 1 Association of average use of 9 a priori text features over a 12-month period with current depression symptom severity.**

| TEXT FEATURE | Example | Words per feature | BETA | SE | P-value |
|---|---|---|---|---|---|
| NEG. EMO. | Hurt, Ugly | 744 | 0.17 | 0.03 | <0.001 |
| 1ST PERS. SING. | Me, I, Mine | 24 | 0.14 | 0.03 | <0.001 |
| 2ND PERS. | You, your | 30 | 0.08 | 0.03 | 0.02 |
| SWEAR | Damn | 131 | 0.11 | 0.03 | <0.001 |
| NEGATIONS | Not, Never | 62 | 0.12 | 0.03 | <0.001 |
| 1ST PERS. PL. | We, Our | 12 | −0.11 | 0.03 | <0.001 |
| ARTICLES | A, The, An | 3 | −0.11 | 0.03 | <0.001 |
| POS. EMO. | Love, Nice | 620 | −0.07 | 0.03 | 0.03 |
| 3RD PERS. | She, They | 28 | 0.06 | 0.03 | 0.06 |

and motivation) has never been formalised and needs considerable further study. With these limitations in mind, we tested if, similar to what has been predicted for self-report symptoms, depression-associated linguistic features are more inter-connected in individuals who have greater self-report depression severity and become more connected, within-subject, during a depressive episode. We first constrained our analysis to networks constructed from 9 text features[26,34,39] that previously studies have linked to depression in the literature to ensure the independence of our analyses, but we subsequently test if our results generalize to a range of networks constructed from depression-associated linguistic more broadly.

In this work, we show that participants with greater depression severity have higher overall network connectivity among a network of 9 a priori selected depression-relevant linguistic features. Among participants with self-reported depressive episodes, we found that network connectivity is higher within vs. outside an episode. These results were not dependent on our chosen network; networks constructed from random samples of depression-related linguistic features are significantly more connected during a depressive compared to networks of depression-irrelevant linguistic features. Our study illustrates that Twitter data, albeit noisy, can be used as an alternative to ecological momentary assessment to study depression longitudinally and in this case, test key predictions of network theory.

## Results

We tested whether global personalised network connectivity, constructed from an a priori set of 9 depression-relevant linguistic features, is associated with baseline depression severity using Twitter data from 946 participants. In a subset of 286 participants, we sought to determine whether within-subject personalised network connectivity is greater within vs. outside a self-reported depressive episode. Finally, we checked whether changes in within-episode network connectivity generalised to 1,000 other combinations of 9-node networks.

**Association of Twitter text features with current depression symptomatology.** As an initial step, we verified whether Linguistic Inquiry and Word Count (LIWC) text features averaged over the past year in our sample were significantly associated with current depression symptom severity (Table 1). Language pertaining to negative emotions (Neg. Emo), use of 1st person singular (1st Per. Sing.), use of 2nd person pronouns (2nd Pers.), swear words (Swear), and negations (Negate) were significantly positively associated with current depression symptom severity. While use of 1st person plural pronouns (1st Pers. Pl.), articles (Articles), and words pertaining to positive emotions (Pos. Emo) were negatively associated with depression severity. There was no

significant association with 3rd person pronouns (3rd Pers.) (Fig. 1). The proportion of days with Tweets within-subject that contained each of the 9 text features is presented in the online supplement (Table S2). Swear words were the least frequently occurring text feature (Mean: 0.30, SD: 0.22), while articles were the most frequent (Mean: 0.80, SD: 0.14).

In terms of consistency with the prior literature, negative emotions[28,34,35,40–42], 1st person singular pronouns[27,28,30,35,36,40,41], swear words[34,36,37,39,40,43], and negations[36,37,40,44] were shown to be positively associated with depression severity. While, positive emotions[31,36,40,43,45,46], 1st person plural[28,36,41], 2nd person[36,40,44], and 3rd person pronouns[36,40,41,44,47] have been found to be negatively associated with depression. Article use has been found to be significantly associated with depression severity, although there is inconsistency regarding the direction of the effect[28,36,37,39,40]. We therefore replicated previously established directional associations for 6 (negative emotions, 1st person singular, 1st person plural pronouns, swear words, negations, and positive emotions) of 9 LIWC text features, but were unable to replicate negative associations for 2nd person (we found a significant positive association) and 3rd person plural pronouns (trend-level in the direction of a positive association). Given conflicting evidence surrounding the direction of associations between article use and depression in the existing literature, our finding of a negative association can be neither confirmatory or dis-confirmatory at this point. Directionality notwithstanding, we were thus broadly assured that the text features we selected showed relevance to depression. Therefore, we used this set of linguistic features as nodes upon which to construct personalised depressive networks.

**Overall depression network composition.** We constructed personalised networks for each participant, based on these 9 depression-associated text features derived from Tweets posted over the 12 months preceding study enrolment. From these individual networks, we tested how network structure differed as a function of depression severity. In support of a central hypothesis of network theory, we observed a significant positive association between depression severity and global network strength ($\beta = 0.008$, SE = 0.003, $p = 0.002$) (Fig. 2a). That is, those individuals with the highest levels of depression had the most tightly connected networks in the sample. Participants with higher depression severity had significantly larger node strength of negative emotions, swear words and articles [Neg. Emo: $\beta = 0.02$, SE = 0.007, $p = 0.007$; Swear: $\beta = 0.02$, SE = 0.007, $p = 0.009$; Articles, $\beta = 0.01$, SE = 0.003, $p < 0.001$] (Fig. 2b). The overall network of depression-related linguistic features was characterised primarily by several weak positive connections, and one strong positive connection between negative emotions and swear words (Fig. 2c). There was a significant positive association between number of days and global network connectivity ($\beta = 0.00009$, SE = 0.00002, $p < 0.001$) (Figure S3a). However, there was no significant association between number of days in the time-series with current depression severity ($\beta = 0.0002$, SE = 0.0003, $p = 0.43$) (Figure S3b) and the significant association with current depression severity remained after controlling for the number of days in each personalised network (Table S3). Furthermore, our findings were not affected when networks were constructed without 3rd person pronouns, which had no significant association with current depression severity (Figure S4).

**Within-subject changes in network connectivity during depressive episodes.** To test the hypothesis that networks of depression-associated linguistic features become more tightly connected during an episode, we compared network connectivity within-subject for periods when participants were depressed

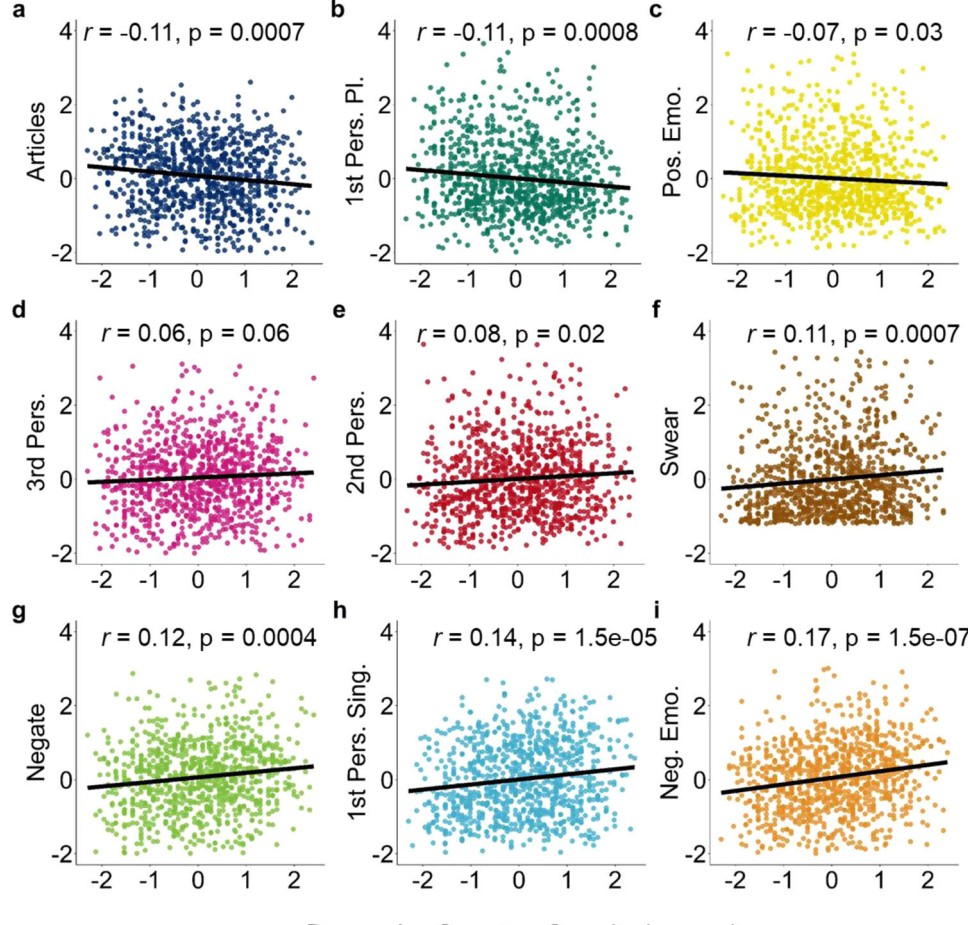

**Fig. 1 Self-reported depression severity is associated with several text features derived from Tweets. a–i** Association between self-report depression symptom severity and the mean of LIWC text features over the past year. There is a significant association between self-reported depression and every LIWC text feature, except 3rd person pronouns **d**. As we decided to construct networks based on these textual features a priori based on the work of de Choudhury et al. (2013), we nonetheless retained 3rd person pronouns in the subsequent analyses. Uncorrected two-sided Pearson correlations. Source data are provided as a Source Data file.

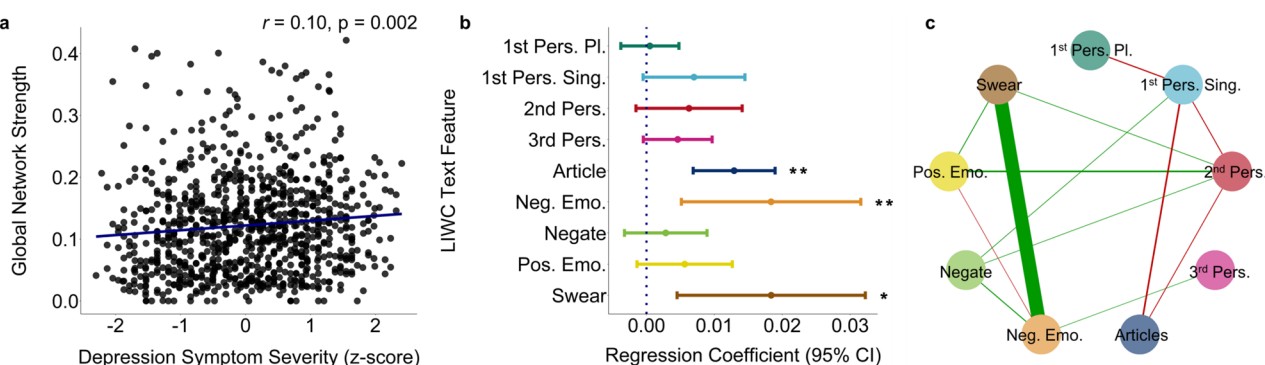

**Fig. 2 The connectivity of personalised networks of depression-relevant language is associated with individual differences in self-reported depression severity. a** There was a significant positive association between global network strength and depression severity ($\beta = 0.008$, SE = 0.003, $p = 0.002$) ($N = 946$). **b** Mean regression coefficients with 95% CIs for individual network node strengths, positive coefficients indicate increased node strength with greater depressive severity. There was a significant association between depression severity and the node strength of Neg. Emo ($\beta = 0.02$, SE = 0.007, $p = 0.007$), Swear ($\beta = 0.02$, SE = 0.007, $p = 0.01$), and Articles ($\beta = 0.01$, SE = 0.003, $p < 0.001$). **c** Mean personalised network of all participants. Green and red lines indicate positive and negative associations respectively. Line widths represent edge strength between nodes, with larger widths corresponding to greater edge strength. Associations were not affected after adjusting for the number of days, a proxy for network stability, (Table S3) nor were they altered by omitting 3rd person pronouns from the network (Figure S4). [a]Unadjusted two-sided Pearson correlation, [b]Two-sided general linear regression unadjusted for multiple comparisons. Source data are provided as a Source Data file. *$p < 0.05$, **$p < 0.01$, ***$p < 0.001$.

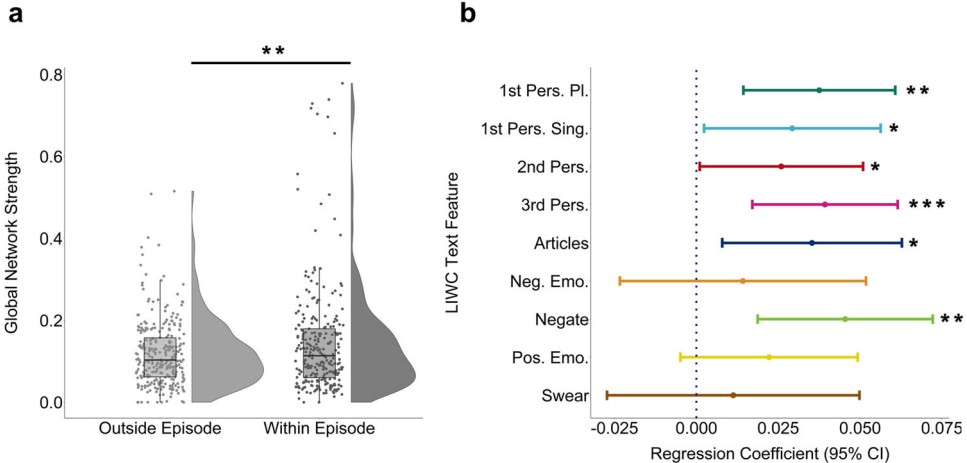

**Fig. 3 Personalised network connectivity increases during a depressive episode for specific symptoms. a** Regression coefficients for global network strength and **b**) mean individual network node strengths with 95% CIs for personalised networks of participants (N = 286) with a depressive episode in the past year for periods within and outside an episode, corrected for number of days. There was a significant increase in global network strength (β = 0.03, SE = 0.009, p = 0.005) and, among individual nodes, a significant increase in 1st person singular (1st Pers. Sing., β = 0.03, SE = 0.01, p = 0.03), 1st person plural (1st Pers. Pl., β = 0.04, SE = 0.01, p = 0.002), 2nd person (2nd Pers.), β = 0.03, SE = 0.01, p = 0.04), 3rd person (3rd Pers., β = 0.04, SE = 0.01, p < 0.001), use of articles (Articles, β = 0.04, SE = 0.01, p = 0.01), and negation words (Negate, β = 0.05, SE = 0.01, p = 0.001) node strength with an episode. Regression coefficients are derived from a within-subject linear model and no adjustments were made for multiple comparisions. The boxplot depicts the median (centre line), upper and lower quartiles, i.e., interquartile range, whiskers, i.e., ±1.5x interquartile range. [a,b]Unadjusted two-sided within-subject regression model. Source data are provided as a Source Data file. *p < 0.05, **p < 0.01, ***p < 0.001.

versus non-depressed over the preceding 12-month period. This required the construction of two personalised networks per person—one 'within episode' and another 'outside episode' among a subset of the sample who reported an episode in the past 12 months (N = 286). In line with our hypothesis, the networks of our participants had a significantly higher global network strength if constructed using language data gathered during an episode ('within') versus a time when they were not currently depressed ('outside') (β = 0.03, SE = 0.009, p = 0.005, Fig. 3a). We found our results were robust to unequal variances in the distribution of global network strength with the Wilcoxon-Signed Rank Test (V = 16,840, p = 0.009). We also performed the analysis using a bootstrapped sample of 80% of the data and re-did the within-subject analysis 1,000 times as a strong control for skewed strength centrality distributions. We found that the distribution of within-episode regression coefficients was significantly above zero (β = 0.03, SE = 0.0001, p < 0.001) (Figure S5). In terms of the specific nodes of the network, during a depressive episode, 1st person singular (1st Pers. Sing., β = 0.03, SE = 0.01, p = 0.03), 1st person plural (1st Pers. Pl., β = 0.04, SE = 0.01, p = 0.002), 2nd person (2nd Pers., β = 0.03, SE = 0.01, p = 0.04), 3rd person (3rd Pers., β = 0.04, SE = 0.01, p < 0.001), use of articles (Articles, β = 0.04, SE = 0.01, p = 0.01), and negation words (Negate, β = 0.05, SE = 0.01, p = 0.001) all had significantly larger node strengths than the same networks constructed during times when the participants were not currently depressed (Fig. 3b).

Changes in node strength were not due to mean increases in the text features themselves, because there were no significant differences among any of the text features within versus outside an episode (Table S4). However, within-and outside episode networks of these participants had an average duration of 80.8 days (SD: 61.7) and 171.5 days (SD: 85.6) respectively, meaning that on-average participants spent considerably more time in a non-depressed state than in a depressed one (β = −90.8, SE = 6.2, p < 0.001, Figure S3c). This gave us cause for concern as we noted a significant negative association between global network connectivity and the number of days that within-

episode networks were based on (r = −0.16(286), p = 0.01) (Figure S3d). The relationship is non-linear with the largest global connectivity values found during short (i.e., under 30 days) within-episode periods despite the removal of outliers. Additionally, there was a significant interaction such that the direction of association between the number of days of data and network strength depended on whether the data was from within vs. outside an episode (β = 0.0005, SE = 0.0001, p < 0.001). We reasoned, therefore, that the difference in number of days upon which within vs outside episode networks were based presented a potential confound to interpretation. Indeed, a permutation test that randomly shuffled the identifier (within-episode with outside-episode) within each participant showed a greater bias towards elevated connectivity for those (fake) within-episode periods than would be expected by chance (β̂ = 0.007, SE = 0.0002, p < 0.001; Figure S6). Importantly, however, 99.3% of the betas observed were smaller than in the real unshuffled data, meaning that over and above any bias introduced by differences in the number of days within/outside episode, the true designation of being within an episode led to higher network connectivity. Consistent with this, after adjusting for the number of days in our regression, the significant increase in global network connectivity within versus outside an episode was reduced, but still statistically significant (β = 0.02, SE = 0.01, p = 0.02). Of the individual node strengths examined, only articles (Articles, β = 0.03, SE = 0.02, p = 0.04) still had a significantly larger node strength within an episode. Thus, the finding of increased network connectivity within versus outside an episode for our a priori depression network survived correction for the number of days within episode.

**Generalisability of findings to other depression networks**. The network of depression-associated features that we constructed was based on features previously described in the literature, and designed to ensure the independence of our analysis from mean-level effects or indeed noise in the present dataset. But it is important to note that this is not the only depression-related

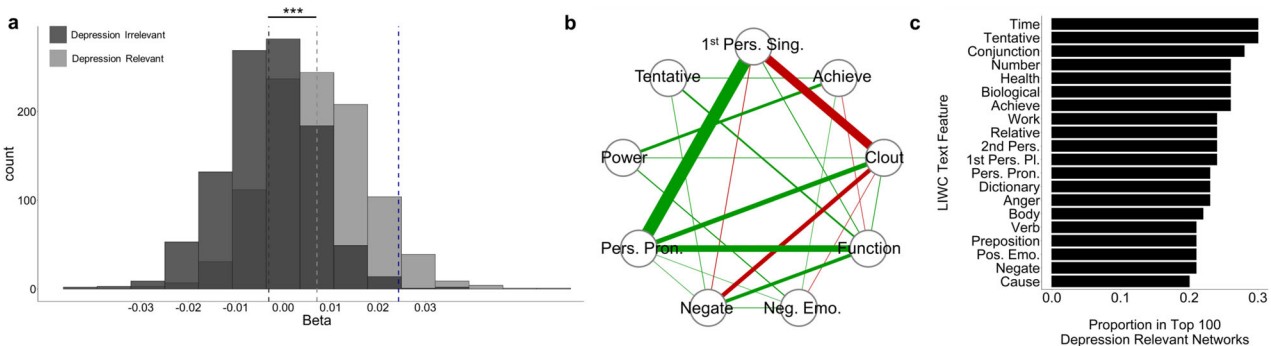

**Fig. 4 The generality of within-subject changes in network connectivity to other depression-relevant networks. a** Effect of episode on global network connectivity in 2000 random networks constructed from text features either significantly (light grey, $N = 1000$) or not significantly (dark grey, $N = 1000$) associated with current depression severity in bivariate correlations at the alpha = 0.05 level. The dark grey and light grey dashed lines indicate the means of the depression irrelevant and relevant distributions respectively. Approximately 59% of LIWC text features were significantly associated with current depression. After adjusting for the number of days, networks of text features associated with depression have a significantly larger regression coefficient indicating elevated within-episode network connectivity ($\beta = 0.01$, SE = 0.0005, p < 2e-16). The effect of within-episode period in our network selected a priori is shown by a dashed blue line. Unadjusted two-sided general linear regression model. **b** For illustration purposes, we show the personalised network with the largest increase in within-episode connectivity ($N = 1$ network). Nodes correspond to 1st person singular pronouns ("1st Pers. Sing."), clout ("Clout": non-transparent summary variable indicating social status/leadership), personal pronouns ("Pers. Pron.", e.g. you, they), function words ("Function" e.g. on, and), tentative ("Tentative", e.g. maybe), negative emotions ("Neg. Emo.", e.g. ugly), power ("Power", e.g. superior), negation ("Negate", e.g. not), and achieve ("Achieve", e.g. win). **c** Text features most likely to appear in networks sensitive to within-subject changes in depression status. Tentative and time related words were found in 30% of the 100 depression relevant networks found to be most sensitive to episode. Source data are provided as a Source Data file. *p < 0.05, **p < 0.01, ***p < 0.001.

linguistic network that can be constructed from these data, nor is it necessarily the best. Of the 87 LIWC text features at our disposal, 59% were significantly associated with current depression severity at an uncorrected $p < 0.05$ level. Bivariate correlations between all LIWC text features and current depression severity can be found in the supplementary material (Table S5). We thus tested if our results held when networks were constructed from different sets of 9 randomly selected text features associated with current depression severity. Networks of text features associated with depression have significantly larger within-episode connectivity than those of networks *not* associated with depression ($\beta = 0.01$, SE = 0.0005, p < 0.001, Fig. 4A). The network with the largest increase in within-episode connectivity (Fig. 4B) included the following depression relevant features: 1st person singular pronouns ("1st Pers. Sing"), clout ("Clout": non-transparent summary variable indicating social status/leadership), personal pronouns ("Pers. Pron.", e.g. you, they), function words ("Function" e.g. on, and), tentative ("Tentative", e.g. maybe), negative emotions ("Neg. Emo.", e.g. ugly), power ("Power", e.g. superior), negation ("Negate", e.g. not), and achieve ("Achieve", e.g. win). In the top 100 depression relevant networks, time and tentative words were found in 30% of networks (Fig. 4C). Our a priori selected network is consequently not the only network with elevated within-episode connectivity nor is it the network with the largest increase in connectivity. But rather is part of a general trend that networks constructed from depression-relevant language features have greater connectivity when in the midst of a depressive episode.

## Discussion

The network theory of mental illness posits that causal interactions between symptoms result in positive feedback loops that lead to the development and maintenance of poor mental health episodes. This theory generates a range of predictions that have been difficult to examine in self-reported depression symptom data due to the difficulty in collecting large volumes of longitudinal self-report data. We adopted an approach to test these

predictions by studying time series of linguistic features that are associated with depression, extracted from the social media platform Twitter. These linguistic features are outwardly observable indicators of a range of internal states that prior work as shown to be relevant to depression. While these linguistic features of depression cannot be directly mapped to individual clinically recognised symptoms, we nonetheless posited that they might interact and serve to reinforce one-another just as has been predicted by network theory for classic symptoms of depression. We predicted that networks constructed from these depression-relevant language features would be more strongly connected in those with higher levels of depression severity and moreover that they would become even more tightly connected when people were in a depressed state.

To test these initial predictions, we took a conservative approach in using 9 a priori text features with previously established relevance to depression from archival Twitter data. We found significant associations between 8 of 9 text features selected and current depression severity, of which 6 were consistent with now well-established directionality in the literature. These included positive associations between the use of 1st person singular pronouns and negative emotions and depression symptom severity. Next, we constructed personalized networks from these 9 features and found that higher levels of current depression severity were associated with greater connectivity of our a priori depression-associated linguistic network across participants. Crucially, we then leveraged the longitudinal nature of this dataset to study how connectivity changes within-subject as their mental health changes. Participants retrospectively reported periods of time when they had a depressive episode in the past year and we constructed networks for 'within' and 'outside' these dates. We demonstrated that the connectivity of depression-related linguistic networks increased within-subject as participants moved into periods of depression. This was true of our a priori network, but crucially also for a range of 9-node networks constructed from randomly selected text features that had related to overall cross-sectional depression symptom severity. That is, networks constructed from depression-relevant text features were

more likely to become tightly connected during an episode than networks constructed from depression-irrelevant text features.

Network theory offers a compelling explanation for the heterogeneity of disorders[48] and are supported by patients' experiences of causal relationships between symptoms[49,50] and the efficacy of cognitive behavioural therapy, which aims in part to diminish associations between symptoms[51]. However, there has been conflicting evidence in the literature regarding whether individuals with a mental illness have greater symptom network connectivity than healthy participants[52]. These results can partially be explained by the over reliance on cross-sectional data, which potentially averages out individual differences in network connectivity. Two prior studies found preliminary within-subject evidence of an increase in network connectivity during an acute phase of mental illness[24,25]. However, both involved only one participant. In this study, we established an increase in within-subject connectivity of a depression-relevant network during a depressive episode in a large sample. Crucially, we also leveraged Twitter text features as a tool for estimating personalised depression networks, rather than using self-report data. While that can be construed as a strength of our investigation, it is also a major weakness; there remains a critical gap in testing if self-reported symptoms would behave in a similar manner.

Using network analysis to understand individual vulnerability to depression is a promising avenue for potentially developing novel and personalized interventions. This is because symptoms with a high strength centrality are thought to have a disproportionate ability to activate or deactivate other symptoms. For example, evidence from a cross-sectional social anxiety disorder network suggests that changes in the most central symptoms in anxiety networks are predictive of more distributed changes in symptoms[53]. In a prospective study of anorexia nervosa patients, higher levels of the most central symptoms at baseline were negatively associated with successful recovery (more so than less central symptoms)[54]. However, a major caveat of research thus far is the use of cross-sectional networks to derive key insights – in some cases these align with personalised networks, but in others not[23]. It is hoped that a push towards the development of more individualised approaches to network estimation will allow us to translate these basic findings into clinical practice. This might take the form of targeting symptoms that are, for the individual, most central, thereby preventing an individual from developing a disorder in the first place[55]. To realise this potential, interpretable depression features will likely be essential. While we believe our data shows an interesting generalisation of network theory beyond self-report symptoms, more work will be needed to extract clinically actionable insights, if they exist, from the study of linguistic features of depression.

Our study was not without limitations and caveats. First, we are not suggesting that Twitter posts will ever (or should ever) be used to make clinical decisions. People on social media tend to selectively express their emotions, i.e. impression management, which obfuscates their true emotional state[56]. Some use these platforms for work, to sell things, for self-promotion and in some cases, to vent their emotions. Therefore, the indirect assessment of depression-relevant language through text analysis will always lead to data that is substantially nosier than otherwise obtained via ESM and would never be of sufficient quality, in our view, to make individualised predictions. Indeed, the effect sizes reported here are low. This is in part because the linguistic features in tweets have low correlations with overall depression severity and also because our definition of a depressive episode was broader than is typical and based on a retrospective report. It therefore remains of key importance to establish if networks of goal-standard assessments of self-report depression symptoms display the same characteristics of the linguistic features studied here and

to establish if effect sizes are clinically meaningful in such datasets. That said, we believe the present findings are of significant theoretical importance in two key ways.

First, in a large enough sample, we can use noisy data like this to test key aspects of network theory. The broad alignment of our findings with the prior literature (e.g., overall association of network connectivity and depression severity) and predictions of network theory (e.g. within subject changes in connectivity during episodes) affirm there is clear signal in these data. There is significant potential, we believe, in using such data to answer questions that are otherwise practically impossible using EMA. Second, the proof of principle established here suggests that other sources of linguistic data that are potentially more indicative of current mood (e.g., text messages, speech) could be mined to help deliver personalized warning signs to individuals. In this context, it is noteworthy to also acknowledge that our ground-truth measure with respect to depressive episodes is a retrospective report and likely subject to errors in recall. This would only serve to diminish our effect size further, which in our view, makes our results more compelling. In a real-world context, it is likely that changes in connectivity associated with episodes of depression are stronger than reported here.

Other limitations to our data include the fact that social media users tend to be younger, better educated, wealthier, and politically more liberal than the general population, posing potential generalizability problems[57,58]. Additionally, online workers on platforms similar to ClickWorker have their own unique socio-demographic profile and crucially, endorse higher rates of a range of mental health problems than the general public[59,60]. In the present study, a large proportion of participants had received a mental health diagnosis in the past and more than half reported a depressive episode in the past year. The high rates of the latter were likely partially inflated by our decision to require just 2 symptoms of depression (low mood and reduced interest) to have been present consistently for a two-week period (instead of the usual 5 of 7)[61], but are likely to be partially due to the known profile of online workers. It remains to be tested if the findings from this study will generalize to individuals recruited via other means and indeed through clinical settings. The majority of text and sentiment analysis libraries used to examine the association between language features and mental health are only available in English[62]. Significant differences have been shown in use of negative, positive emotions, personal pronouns, articles, and other lexical attributes between social media users in western (U.S. and U.K.) and non-western (India and South Africa) countries[63]. The vast majority of our sample came from predominantly English-speaking countries. Because of this, we do not how these associations may generalise to different languages and cultural settings. Similar to several other personalised network papers[64,65], we found that none of the nodes in our a priori network was normally distributed. Network analysis assumes that all nodes are multivariate normally distributed[66], however it is not yet known the extent to which edge and centrality estimates are affected by deviations from normality. Finally, the LIWC is only able to account for the proportion of words in a particular category, e.g., proportion of $1^{st}$ person pronouns in a passage of text. Any context or more subtle usage of language, such as irony, that would change the underlying emotional meaning of a text are not captured by this method. More broadly, there are a range of more sophisticated analytical approaches when it comes to the content and sentiments in tweets that may prove stronger indicators of depression (but see[67]) and thus better candidate nodes to construct depression-relevant networks. We chose instead to use an established library and to focus on language features previously shown to relate to depression to keep a degree of independence in the datasets used to derive depression-relevant

**Table 2 Demographic and Twitter use characteristics of sample.**

| | Full Sample N = 946 | No Depressive Episode N = 388 | Depressive Episode N = 558 | p-value |
|---|---|---|---|---|
| Twitter Behaviour | | | | |
| Tweets | 356.1 (581.4) | 306.1 (517.2) | 390.8 (620.2) | 0.03* |
| Retweets | 276.0 (494.1) | 246.7 (468.0) | 296.5 (510.8) | 0.13 |
| Likes | 1097.7 (1062.7) | 983.4 (996.6) | 1177.2 (1100.2) | 0.006** |
| Recruitment Type (% Paid) | 680 (71.9%) | 274 (70.6%) | 406 (72.8%) | 0.52[a] |
| Word count per day | 131.6 (127.7) | 115.2 (114.8) | 143.1 (134.9) | 0.001** |
| Age (years) | 29.6 (10.6) | 30.8 (11.3) | 28.7 (10.1) | 0.002 |
| Gender | | | | |
| Male | 304 (32.1%) | 147 (37.9%) | 157 (28.1%) | <0.001***[a] |
| Female | 617 (65.2%) | 240 (61.9%) | 377 (67.6%) | – |
| Transgender Male | 6 (0.6%) | 0 (0%) | 6 (1.1%) | – |
| Transgender Female | 1 (0.1%) | 1 (0.3%) | 0 (0%) | – |
| Non-Binary | 15 (1.6%) | 0 (0%) | 15 (2.7%) | – |
| Other | 3 (0.3%) | 0 (0%) | 3 (0.5%) | – |
| Country | | | | |
| Ireland | 44 (4.7%) | 24 (6.2%) | 20 (3.4%) | 0.15[a] |
| United Kingdom | 339 (35.8%) | 147 (37.9%) | 192 (34.4%) | – |
| United States | 480 (50.7%) | 188 (48.5%) | 292 (52.3%) | – |
| Canada | 42 (4.4%) | 13 (3.4%) | 29 (5.2%) | – |
| Australia | 15 (1.6%) | 4 (1.0%) | 11 (2.0%) | – |
| Other | 25 (2.6%) | 12 (3.1%) | 14 (2.5%) | – |
| Education | | | | |
| Less than high school | 16 (1.7%) | 4 (1.0%) | 12 (2.2%) | <0.001***[a] |
| High school | 190 (20.1%) | 66 (17.0%) | 124 (22.2%) | |
| Some university | 317 (33.5%) | 108 (27.8%) | 209 (37.5%) | – |
| Bachelor's degree | 285 (30.1%) | 142 (36.6%) | 143 (25.6%) | – |
| Master's degree | 104 (11.0%) | 53 (13.7%) | 51 (9.1%) | – |
| Professional degree | 15 (1.6%) | 4 (1.0%) | 11 (2.0%) | – |
| Doctorate | 19 (2.0%) | 11 (2.8%) | 8 (1.4%) | – |
| Currently Employed (% Yes) | 458 (48.4%) | 204 (52.6%) | 254 (45.5%) | 0.04*[a] |
| Physician diagnosed depression (% Yes) | 432 (45.7%) | 76 (19.6%) | 356 (63.8%) | <0.001***[a] |

Twitter and demographic characteristics of all participants along with differences between participants with and without a depressive episode.
*p < 0.05, **p < 0.01, ***p < 0.001, [a]Chi-square test.

features and the one (here) used to study how their network compositions changes through time. Future work might draw on alternative methods and have greater power to interrogate network dynamics in these datasets.

We found support for two of the principal predictions of network theory using a proxy for longitudinal (historical) EMA. Specifically, we found that the connectivity (partial correlation) between a set of pre-defined linguistic features of depression relates to an individual's current depression symptom severity. Moreover, we found that this network connectivity increases within-subject during a depressive episode. Future work can utilize this methodological approach to test and refine key aspects of network theory. Elevated network connectivity within an episode was not specific to the a priori LIWC text features chosen; they generalised to a broader set of linguistic features that are associated with depression severity and future research might elect to utilize the best performing network we identified here. Whether these findings generalise to other aspects of mental health is not yet known. Recent work suggests that there are a host of commonalities across various aspects of mental health in their use of language on Twitter. Regardless of whether there is some degree of specificity of the nodes that comprise such networks, it will be interesting to determine if network connectivity increases, within-subject, occur during the acute phase of other mental health illnesses, such as bipolar disorder. Given the vast amount of data available and its longitudinal archival nature, social media network analysis is a promising method for testing some of the tougher predictions of network theory, albeit using very different 'markers' of depression.

## Methods

**Participants**. We recruited 1,713 participants for this study. The majority were recruited on Clickworker (N = 1,395), an online worker platform, and were paid €2.5 for their participation. A smaller number participated voluntarily (i.e. without payment) and were recruited through general advertising on Twitter and in print media (N = 318). Participants were included for analysis if they were at least 18 years old and had a Twitter account with at least 30 days of tweets and if at least 50% of their tweets were in English. They were also required to pass an attention check, a combination of a captcha and an item with an obvious correct response ("Please select 'A little' if you are paying attention"). Of the 1,713 participants recruited, 99 were excluded due to failing the attention check and a further 668 participants were excluded for either not having at least 30 days of tweets or fewer than 50% of their tweets were in English. After excluding these participants, 946 participants were brought forward for analysis. Participants had a mean age of 29.6 years (SD: 10.6, range: 18-66), a majority were female (65.2%), currently unemployed (51.6%), and resided in either the U.K. (35.9%) or U.S. (50.7%).

Particpants reported more than half (59.0%) of the sample reported at least one depressive episode in the past year (mean: 1.56 episodes, SD: 0.81) with an average duration of 104.06 days (SD: 97.06) and 45.7% reported being diagnosed by a physician with depression at some point in their life. Participants were asked to self-report the dates of any depressive episodes in the past year; a depressive episode was defined as a period of at least two weeks with low mood and loss of interest or pleasure in activities every day or nearly every day. Participants that reported at least one depressive episode tweeted (β = 84.7, SE = 38.4, p = 0.03) and liked the Tweets of others (β = 193.8, SE = 70.0, p = 0.006) more frequently than participants without a depressive episode, but did not retweet significantly more (β = 49.8, SE = 32.6, p = 0.13). Individuals who reported a depressive episode in the past 12 months were younger (β = −2.2, SE = 0.70, p = 0.002), more female ($\chi^2$(5, N = 946) = 26.0, p < 0.001), less likely to be employed ($\chi^2$(1, N = 946) = 4.3, p = 0.04), and more likely to have been diagnosed with depression by a physician ($\chi^2$(1, N = 946) = 178.5, p < 0.001). Individuals that reported a depressive episode were also more likely to have a lower educational attainment than those without a depressive episode ($\chi^2$(6, N = 946) = 28.0, p < 0.001). There was no significant difference in country of residence for individuals with versus without a depressive episode ($\chi^2$(5, N = 946) = 8.1, p = 0.15) (Table 2). Participants recruited through

Clickworker tweeted ($\beta = -217.8$, SE = 41.5, $p < 0.001$), retweeted ($\beta = -162.0$, SE = 35.4, $p < 0.001$), and liked other posts ($\beta = -227.0$, SE = 76.5, $p = 0.003$) significantly less than people who were not paid for their participation (Table S1). Furthermore, although there was no difference between groups in the percentage of depressive episodes in the past year ($\chi^2(1, N = 946) = 0.42$, $p = 0.52$), paid participants were significantly less likely to have been ever been diagnosed by a physician with depression ($\chi^2(1, N = 946) = 6.9$, $p = 0.01$).

**Procedure.** After providing informed consent, participants were asked to complete a self-report questionnaire and provide their Twitter handle which was used to collect the most recent (max 3,200) tweets and (max 3,200) likes from their account. Tweets were collected using a data collection app written in Python using the Twitter developer's Application Programming Interface. Participants were asked to provide their age, gender, country of residence, current employment status, and highest educational attainment. They were also asked if they have ever been diagnosed by a physician with depression and if yes to provide the approximate date of diagnosis. Next, they completed a self-report depression questionnaire to establish their current symptom severity levels. In the first wave of recruitment, 263 participants completed the Centers for Epidemiologic Studies Depression scale[68] (CES-D 8). In subsequent recruitment waves, the remaining 1,450 participants completed the Zung Self-Rating Depression Scale (SDS)[69] instead. We combined scores from the two depression scales by standardizing each scale by its mean and standard deviation. Finally, participants were asked to report up to five depressive episodes in the past year. A depressive episode was defined as a period of at least two weeks in which the participant felt both low mood and loss of interest or pleasure in hobbies and activities nearly every day for most of the day. We chose this definition to increase the sensitivity for detecting depressive episodes and to reduce participant burden by only requiring the two essential components of a depression diagnosis. Episodes were recoded to be "not depressed" if they were shorter than 2 weeks in duration and were merged together if separated by fewer than 2 weeks (effectively recoding intervening days as also being depressed).

**Pre-processing and text analysis.** We restricted our analysis to tweets published in the 12 months prior to survey completion. Before text analysis, extraneous information was removed from tweets including: reply symbol (@), hashtag symbol (#), emojis, punctuation, links (URLs), and all other non-alphanumeric characters. Periods, exclamation points, and question marks were the only punctuation retained because they are necessary to calculate the number of words per sentence. Tweets were aggregated into daily bins and text analysis was then performed on all tweets published per day per user. Daily observations were chosen to increase the amount of text for reliable estimation of text features. Text analysis of daily Tweets was carried out using the Linguistic Inquiry and Word Count (LIWC 2015) dictionary[70]. The LIWC is a dictionary comprised of approximately 6,400 words and word-stems with 90 different output variables including: linguistic characteristics (e.g. articles and pronouns), psychological constructs (e.g. sadness and positive emotions), and general text information (e.g. punctuation and word count). The LIWC has been used in prior studies that reported a relationship between Twitter sentiments, text features, and depression[26,39]. As an initial step to verify that these features were picking up depression symptomology, we averaged each feature over the past 12 months and then tested for correlation with current depression severity.

Among the 9 averaged LIWC text features, any value more than 3 standard deviations from the group mean for that text feature was subsequently removed. Approximately 1.1% of data in the full sample of participants was excluded using this criterion.

**Feature specification.** We selected 9 LIWC text features a priori for network analysis based primarily (but not wholly) on the findings of de Choudhury et al. (2013), who found that the following text features had relevance to self-reported depression severity: 1st person singular ("1st Pers. Sing.", 24 words incl. "I", "me", "mine"), 1st person plural ("1st Pers. Pl.": 12 words incl. "we", "our"), 2nd person ("2nd Pers.", 30 words incl. "you", "your"), and 3rd person ("3rd Pers.", 28 words incl. "she", "they") pronouns, negative ("Neg. Emo.": 744 words incl. "hurt", "ugly") and positive emotions ("Pos. Emo": 620 words incl. "love", "nice"), swear ("Swear": 131 words incl. "damn"), articles ("Articles": 3 words: "a", "an", "the"), and negation ("Negate", 62 words, incl. "not", "never") words. Specifically, these words were found to have either a significant change in mean, variance, momentum, or entropy in their sample at a stringent correction for multiple comparisons. Based on findings from prior work, however, we did not average the two 1st person pronouns plural and singular together into a single 1st person pronoun category and indeed, prior work has shown they have bidirectional associations to depression and indeed, we confirmed this in our data as well[28,36,41]. Using the LIWC, we calculated the proportion of text on each day with tweets in the past year that included words from each of the LIWC's 87 categories. This resulted in a time-series for each of the 87 text feature categories for each participant. Days without tweets were not considered or assigned any value and consequently participants who tweet less often had fewer days in their time series than participants who tweeted every day.

**Network analysis.** Networks were constructed by examining the correlation between these text feature time series (nodes), using regularized partial correlations to determine the contemporaneous association between text features[71]. The contemporaneous association is based on the residuals of the lag-1 correlation and removes any temporal effects due to other variables measured at the same time point[22,72]. Individual node strength, the sum of the absolute values of partial correlations into a node, and global network strength, average of node strength across all nodes, are the primary indicators of network connectivity in psychological networks. Personalised networks were estimated for each participant using the *graphicalVar* (version 0.2.4) package with LASSO regularisation. Regularized partial correlations control for associations between all other nodes in a network with high specificity. Consequently, an edge that is present in a regularized network likely presents a true edge rather than a false positive. However, as we were not focused on particular edges between nodes, but rather the broader characterisation of 'connectedness', we set the hyperparameter (gamma) to 0, which, although still regularised, causes the model to prefer *more* connections over fewer. This avoided a situation where many edges would be returned as 0 and is the same approach applied in a prior study using cross-sectional networks[11]. A range of 10 tuning parameters (lambdas) was considered for each person's model (nLambda = 10).

**Network Connectivity of a priori Network and Current Depression.** Using this method, we first estimated personalised networks for all participants (N = 946) and created a mean of these personalised networks to describe the network's overall composition regarding strength, closeness, and betweenness centrality. Any individual node strength value, among the 9 LIWC text features, that was greater than 3 standard deviations from that node's group mean was excluded from analysis. Using this exclusion criterion, approximately 2.5% of all node strength values were omitted. In order to test for the reliability of edge strengths in the network, we split our sample into two equal halves, calculated personalised networks for all participants, and then correlated the mean edge strengths between the two halves. We found that among the 36 unique edges in the network, there was a high degree of reliability between the split halves (r(36) = 0.99, p < 0.001). The edge strength between Neg. Emo. and Swear was much stronger than between other edges, reliability was r(35) = 0.97, p < 0.001 when we exclude this edge. Because individual networks tended to be sparse, the average of most edges tended towards zero leading to a high correlation between split halves (Figure S1a). Global network strength was not normally distributed (Shapiro-Wilk test, W = 0.95, p < 0.001) and had a rightward skew (Figure S1b). The strength centralities of most nodes, expect Swear and Neg. Emo. had a strong right skew due to the relative sparsity of those nodes (Figure S1c). Note we did not calculate closeness and betweenness centralities for individual networks due to edge sparsity and the strong correlation with strength centrality. To test if network connectivity based on the entire 12-month dataset was related to depression symptom severity, we correlated each individual's network characteristics (i.e., network node strengths) with their current depression severity.

**Change to a priori network within vs outside depressive episodes.** Among participants who reported a depressive episode, we estimated two separate personalised networks for each person, representing periods when they were within and outside a depressive episode, hypothesising that networks would be more tightly connected within compared to outside of an acute episode. Individuals were required to have at least 15 days of tweets both within and outside a depressive episode. We compared network connectivity using regression with "depressive episode" (1 = within episode, 0 = outside episode) as a within-subjects variable predicting node strength. We did the same analysis predicting global network strength.

**Stability checks.** Network analysis is made more robust by having fewer nodes and this was why the networks presented here are limited to 9 nodes[73]. The number of nodes in the current study is well within the typical range (5-11) included in other network papers on depression[12,18,20,21,24]. We quantified network stability for each personalised network using the *bootnet* package (version 1.5). Stability was assessed by repeatedly dropping up to 75% of cases in the sample and correlating the resultant strength centrality to the estimate based on the full sample. Stability was quantified with the correlation stability coefficient (CS coefficient), with 1,000 bootstrapped samples, i.e., the maximum proportion of the sample which can be dropped to retain a 0.7 correlation with the full sample in 95% of cases. A simulation study by Epskamp, Borsboom, and Fried (2018) proposed using a threshold for CS coefficients above 0.50 to ensure that the ordering of centralities is interpretable. Personalised networks of all participants were found to be highly stable (mean strength CS coefficient: 0.65, SD: 0.12) along with networks constructed from within (mean strength CS coefficient: 0.50, SD: 0.23) and outside (mean strength CS coefficient: 0.64, SD: 0.13) a depressive episode.

**Generalisability of findings to other depression networks.** We based our initial analysis on an a priori network of text features previously linked to depression. The idea behind this was to be as conservative as possible and ensure independence of the selection of features from this dataset. Following this proof of principle, we tested if this finding would extend to other depression-associated linguistic networks (i.e. networks constructed from other text features that were significantly

associated with depression severity). Crucially, this analysis controls for the possibility that networks of any kind would be more strongly connected within vs outside an episode. We thus hypothesised that networks comprising text features significantly associated with depression would show greater within subject changes in connectivity compared to those not associated with depression. To test this, we constructed a list of all depression-relevant and 1000 depression-irrelevant text features from LIWC, based on an arbitrary threshold of $p < 0.05$ for their bivariate association with depression severity (see supplement Figure S5 for full list of associations). We then selected 1000 random sets of 9 features from each of the depression-relevant and irrelevant lists and estimated personalised networks for all 2000 sets. We did this twice for each particpant, once based on Tweets that were published within an episode and once based on Tweets outside an episode. This allowed us to test if global network connectivity was greater within versus outside an episode for each of the 2000 networks, using regression with "depressive episode" (1 = within episode, 0 = outside episode) as a within-subjects variable predicting node strength (exactly the same analysis as for the a priori network). Finally, we took the 2000 betas for the within-subjects variable 'depressive episode' in these analyses (Fig. 4a: 1000 depression-relevant and 1000 depression-irrelevant betas) forward to a general linear regression to determine if the extent to which episodes became more connected during an episode was greater in depression-relevant compared to depression-irrelevant networks of language use.

**Control analyses**. A range of control analyses are presented in the online supplement. These examine several potential confounding influences in network estimation. First, we noted that participants had more data (i.e., more days) outside an episode than within one. To control for the possibility that differences in the number of days might be driving our results, we (i) conducted a permutation test that randomised the identifier "within episode" versus "outside episode" 1000 times within subject and (ii) ran an additional within-subjects regression analysis that included the number of days within and outside an episode as a covariate. Because 3rd person pronouns (she/he, they) were selected a priori, but not significantly associated with current depression severity in this sample, we repeated our analyses omitting this node to ensure our results were not affected by its inclusion. In the LIWC library, certain 'supra-categories' are inclusive of multiple sub-categories. For example, within the personal pronoun category are: 1st person singular/plural, 2nd person, 3rd person, and impersonal pronouns. To test if this had a material effect on our results, we repeated our analyses excluding these supra-categories and the results were unchanged (Figure S2).

Mean personalised networks were visualised using the qgraph package (version 1.6.9). Between and within-subjects regression were preformed using the glm (version 3.6.1) and lmer packages (version 3.1-3). All statistical analyses were performed in R (3.6.1).

**Reporting summary**. Further information on research design is available in the Nature Research Reporting Summary linked to this article.

## Data availability
The raw Twitter data are protected and are not available due to data privacy. The processed Twitter data are available from the corresponding author on reasonable request. Processed Twitter data cannot be shared due to the possibility of participant identification. The processed data used to generate figures and tables in this study are provided in the Source Data file. Source data are provided with this paper.

## Code availability
The code used to analyse the data in the current study is available at: https://doi.org/10.5281/zenodo.5745764[74].

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

## Acknowledgements

This study was funded by the 'Institutional Strategic Support Fund' grant (204814/Z/16/A) to Trinity College Dublin, funded by the SFI-HRB-Wellcome Trust partnership. SWK is funded by a Provost PhD Project Award awarded to CMG. We would like to thank Caoimhe Ni Mhaonaigh and Louise Burke for their assistance in participant recruitment.

## Author contributions

S.W.K.: Conceptualization, methodology, study design, statistical analysis, writing (drafting and editing). C.M.G.: Conceptualization, methodology, study design, statistical analysis, writing (drafting, editing, and supervision).

## Competing interests

The authors declare no competing interests.
