## [Peer Review File · Nature Communications]

Using linguistic features in social media posts to study the network dynamics of depression longitudinallyEditorial Note: This manuscript has been previously reviewed at another journal that is not operating a transparent peer review scheme. This document only contains reviewer comments and rebuttal letters for versions considered at *Nature Communications*.

REVIEWERS' COMMENTS

Reviewer #2 (Remarks to the Author):

As previously indicated the authors have adequately addressed my concerns with respect to the terminological issue that I pointed out earlier. I also pointed out two very relevant and recent publications in my review that the authors should discuss as previous research, namely: doi.org/10.1038/s41598-020-74314-3 and <https://doi.org/10.1038/s41562-021-01050-7>. These recent papers are quite relevant to the manuscript's discussion of existing studies because they pertain to novel lexical or linguistic features associated with depression in the context of online Twitter data (cf. references 25-35). I recommend the paper be accepted for publication given this minor change.

Reviewer #4 (Remarks to the Author):

I think the authors have responded well to (my single) comment. They have explained their reasoning well and highlighted the outstanding questions clearly. Really interesting paper.

Reviewer #2 (Remarks to the Author):

As previously indicated the authors have adequately addressed my concerns with respect to the terminological issue that I pointed out earlier. I also pointed out two very relevant and recent publications in my review that the authors should discuss as previous research, namely:

doi.org/10.1038/s41598-020-74314-3 and <https://doi.org/10.1038/s41562-021-01050-7>

These recent papers are quite relevant to the manuscript's discussion of existing studies because they pertain to novel lexical or linguistic features associated with depression in the context of online Twitter data (cf. references 25-35). I recommend the paper be accepted for publication given this minor change.

We'd like to thank the reviewer for pointing out these two relevant publications. We have added references to these papers in the introduction section discussion of the prior literature.

Introduction, page 6

Along with changes in pronoun use, depression is associated with negatively biased cognitive distortions, e.g., everyone thinks that I am a loser.[33]

Introduction, page 6

“People with depression are also less active on Twitter in the early morning (3am-6am) than healthy controls exhibiting an altered circadian rhythm, but also used significantly more personal pronouns, negative affect words, and rumination words during this time. Language usage on social media is thus not static, instead changing over time reflecting underlying changes in a participant’s mental health [38].

Reviewer #4 (Remarks to the Author):

I think the authors have responded well to (my single) comment. They have explained their reasoning well and highlighted the outstanding questions clearly. Really interesting paper.

Thank you.